# Cool Mix Asphalt—Redefining Warm Mix Asphalt with Implementations in Korea, Italy and Vietnam

Hosin (David) Lee [1,*], Lan Ngoc Nguyen [2], Elena Sturlini [3] and Young-ik Kim [4]

1   Civil and Environmental Engineering, Iowa Technology Institute, The University of Iowa, Iowa City, IA 52027, USA
2   Department of Construction Materials, University of Transport and Communications, No. 03 Cau Giay, Hanoi 100000, Vietnam; nguyennngoclan@utc.edu.vn
3   Bindi S.P.A, Via Ponte alle Forche, 52027 San Giovanni Valdarno, Italy; e.sturlini@bindistrade.it
4   Hansoo Natech Co., Ltd., 6-20 Yuseong-daero 1205beon-gil, Yuseong-gu, Daejeon 34104, Republic of Korea; hsr6890-ab@ihansoo.com
*   Correspondence: hosin-lee@uiowa.edu

**Abstract:** During the past decades, to minimize Greenhouse Gas (GHG) emissions and asphalt fumes during the asphalt mix production and construction process, various warm mix asphalt (WMA) additives have been developed and successfully applied. Currently, as production of WMA reaches close to that of Hot Mix Asphalt (HMA) in the US, the varied definition of WMA is questioned in this paper. Not only are the temperature reduction ranges from HMA defined by various studies too wide, but also the minimum threshold to be classified as WMA is often too small. In this paper, a new category of "Cool Mix Asphalt (CMA)" is proposed to distinguish it from the newly defined WMA based not on the reduction amount from HMA temperature but its actual production temperature. It is proposed that HMA should be defined as asphalt mixtures produced at temperatures between 140 and 160 °C (between 284 and 320 °F), WMA as production temperatures between 120 and 140 °C (between 248 and 284 °F), and CMA as production temperatures between 100 and 120 °C (212 to 248 °F). By defining their actual production temperatures rather than reduction temperatures from HMA, WMA and CMA will be clearly defined. This paper then presents a new Polymer Cool Mix Asphalt (PCMA) additive called "Zero-M", which was developed to lower the mixing temperature to around 110 °C (203 °F). Recently, test sections using Zero-M were successfully constructed in Korea, Italy and Vietnam, and their laboratory test results of field cores and production and construction experiences are described in this paper. The chemistry and compositions of Zero-M are discussed along with its mechanism to significantly lower the production temperature of PCMA. All test sections constructed in three countries met the in-place compaction density requirements of their respective countries, which were close to or higher than those of the control HMA test sections.

**Keywords:** Cool Mix Asphalt; Warm Mix Asphalt; Hot Mix Asphalt; Polymer Cool Mix Asphalt; Field Density; Implementations in Korea Italy Vietnam; Asphalt Production Temperature

## 1. Introduction

During the past century, the asphalt industry has been always concerned about keeping the temperature of asphalt mix high enough for adequate coating, placement and compaction. An answer to the problems in coating and compaction was to raise the temperature of asphalt mix. Now, for a better performance and cleaner environment, a new



approach is to lower the temperature of asphalt mix, called "Warm Mix Asphalt (WMA)" and "Cool Mix Asphalt (CMA)", which is produced at a lower temperature than WMA [1]. This paper is an expanded version of the keynote paper presented at the 8th International Conference on Bituminous Mixtures and Pavements.

WMA encompasses a range of technologies used to reduce the production temperature of asphalt mixtures [2]. WMA was first introduced in Europe in 1997 and transferred to the U.S. in 2008 through the international technology scanning program [3]. Since that time, the U.S. has been touted as the leading adopter and proponent of the WMA technology. US Governments currently encourage the use of WMA with equal or better performance.

In 2008, the first international WMA conference was held in Nashville, Tennessee. Over 700 pavement professionals from around the world attended the conference. In 2010, FHWA named WMA as an "Every Day Counts" initiative because it was a proven technology with environmental and construction benefits. In 2011, the second international WMA conference was held in St. Louis, Missouri. In 2013, the first Global Warm Mix Asphalt Workshop was held in Iowa City, Iowa. In 2013, The National Asphalt Pavement Association (NAPA) received the NOVA Award from the Construction Innovation Forum for its work to speed the deployment and uptake of WMA [4].

WMA has become a mainstream of the asphalt pavement construction in the United States and the world due to its reduced fuel consumption, lower carbon dioxide emissions, reduced oxidation of asphalt, early opening to traffic and better working environment for workers [5]. However, the plant mixing temperature of WMA varies widely. Typically, WMA is defined as the level of temperature reduction from HMA and, depending on the literature, this ranges between as low as 30 °F (17 °C) and as high as 120 °F (67 °C) lower than conventional HMA technologies [6]. Based on an extensive literature review, the production temperature of WMA is 10–40 °C lower than the conventional HMA [2].

The WMA definition, however, further varies because the temperature of HMA varies between 127 and 168 °C depending on the Performance Grade (PG) binder grade. European Asphalt Pavement Association (EAPA) defines WMA as between 100 and 150 °C and HMA as 120 and 190 °C with some overlapping temperatures between them [7]. In Vietnam, WMA with a 1.5% Sasobit additive and 40% Reclaimed Asphalt Pavement (RAP) was produced at 140 °C and compacted at 125 °C and evaluated for rutting and cracking performance [8]. In Korea, WMA mixtures with 50% Reclaimed Asphalt Pavement (RAP) and 0.2% diamine/fatty acid amine additive were produced at 140 °C and compacted at 120 °C, and the field cores exhibited 4.1% air voids [9]. Mixing and compaction temperatures are key control indicators for construction of asphalt mixtures, but there is no unified standard method that has been established yet [10].

There are two main purposes of this paper: (1) clearly defining HMA, WMA and "Cool Mix Asphalt (CMA)" by their production temperature ranges and (2) demonstrating the successful field implementations of "Polymer Cool Mix Asphalt (PCMA)" using a "Zero-M" additive in Korea, Italy and Vietnam.

## 2. Redefining Warm Mix Asphalt

According to the Iowa Department of Transportation (IDOT) specification, temperatures of HMA, WMA and Polymer-Modified Asphalt (PMA) mixes are defined as follows [11]:

a.  The asphalt binder shall be brought to a temperature of 260 to 330 °F (125 to 165 °C).
b.  The temperature of the HMA mixtures shall not exceed 330 °F (165 °C) unless approved by the engineer.
c.  Keep the production temperature of HMA mixtures between 225 and 330 °F (107 to 165 °C) until placed on the grade.

d. Do not discharge HMA into the hopper when its temperature is less than (1) 245 °F (118 °C) for a nominal layer thickness of 1.5 inches or less, or (2) 225 °F (107 °C) for a nominal layer thickness of more than 1.5 inches.

e. WMA refers to asphalt concrete mixtures produced at temperatures approximately 50 °F (28 °C) or more below those typically used in production of HMA.

f. Keep the production temperature of WMA mixtures between 215 °F and 280 °F (102 to 138 °C) until placed on the grade.

g. The maximum production temperature for WMA is 330 °F (165 °C).

h. Modified asphalt binder should be heated according to the suppliers' recommendations.

Based on the definition of WMA by Iowa DOT in item (f) above, the maximum temperature of the WMA should be 280 °F (138 °C). However, the Iowa DOT specifies that the maximum mixing temperature for WMA is as 330 °F (165 °C) in item (g), which is the same as HMA in items (b) and (c). Since the maximum temperature for PMA binder for HMA is not defined in (h), PMA binder for WMA cannot be determined. Because compaction temperature varies even more, in this paper, the compaction temperatures are not considered to define HMA, WMA and CMA.

As shown in Figure 1, according to NAPA survey data, the usage of WMA mixtures in the USA grew from 69 million tons in 2013 to 169 million tons in 2019, accounting for nearly 40% of the asphalt mixtures' consumption whereas WMA usage in Europe remains low under 10 million tons [12]. Figure 2 shows market shares of different WMA technologies in the USA from 2009 to 2020, where, recently, the use of foaming technology has been decreasing while chemical additives have increased [12]. In the early years, many asphalt mix producers purchased foaming equipment that could be attached to their existing drum mixer, but recently, a significant portion of them stopped using their foaming equipment. The market share of chemical additives has increased due to one pre-mix type product that was added to asphalt and was often supplied as a compaction aid. As can be seen in Figure 2, organic and water-bearing additives have never had a significant market share.

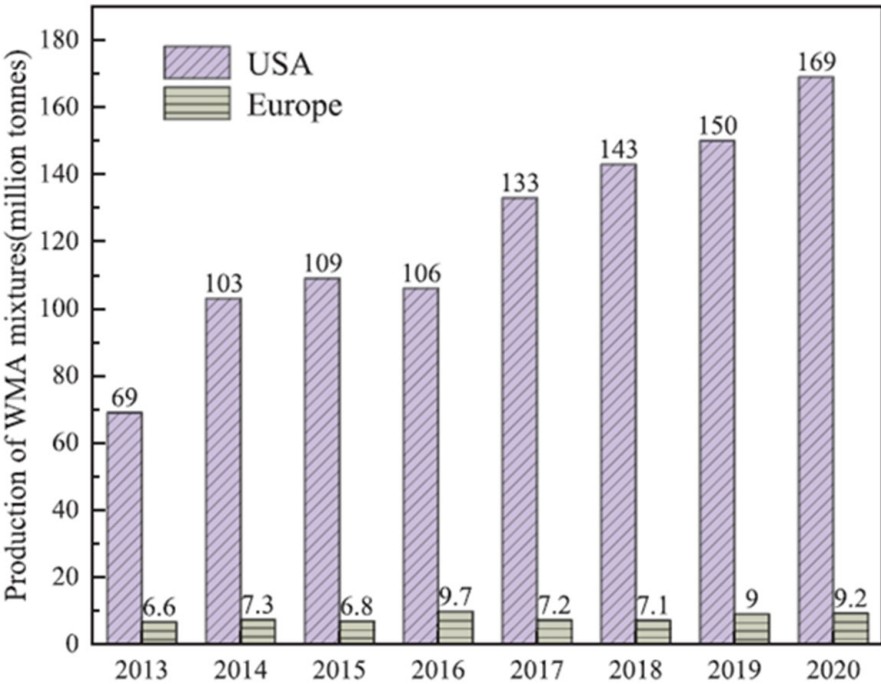

**Figure 1.** Production of WMA mixtures in USA and Europe in 2013–2020 [12].

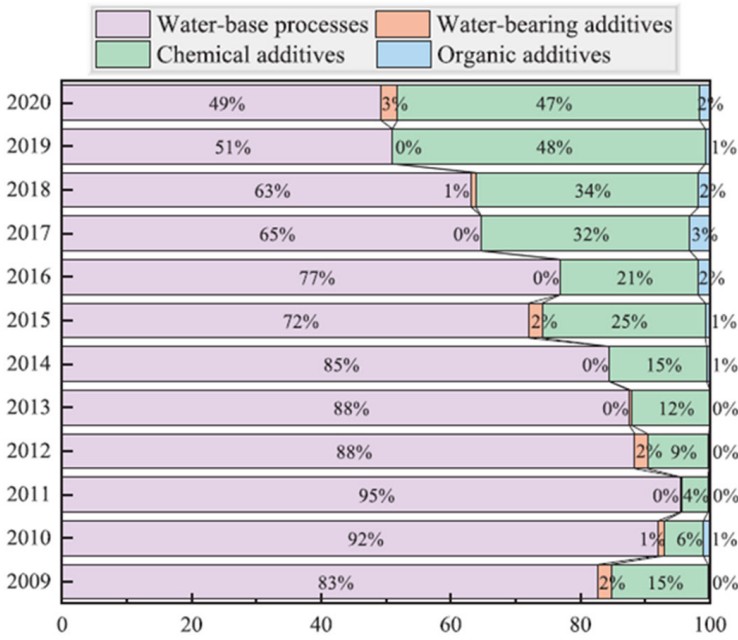

**Figure 2.** Percent production of WMA technologies in the USA [12].

The survey performed by NAPA that generated Figures 1 and 2 provides the following definition: "Warm-mix asphalt is the generic term for a variety of technologies that allow the production of asphalt pavement material to lower the temperature at which the material is mixed and placed on the road by 10 to 100 degrees F (6 to 38 °C)". This definition of WMA, however, is concerning because contractors can claim they are producing WMA with a production temperature reduction of just 10 °F (6 °C), which is too small. With this wide range of temperature definitions for WMA, the survey results shown in Figures 1 and 2 may not be reliable. As discussed earlier, without defining the temperature range for HMA, a 10 to 100 °F reduction in temperature for WMA could create a wider range of plant mixing temperatures of WMA.

It is proposed in this paper that HMA should be defined as asphalt mixtures produced at temperatures between 140 and 160 °C (between 284 and 320 °F), WMA as production temperatures between 120 and 140 °C (between 248 and 284 °F), and CMA as production temperatures between 100 and 120 °C (212 to 248 °F). It may be argued that temperatures between 100 and 120 °C are not a cool temperature, but the same argument can be made for WMA, that temperatures between 120 and 140 °C (or a production temperature reduction of just 6 °C from HMA) are not warm either, but it has been already accepted as an industry standard. Therefore, it is necessary to create a new category of asphalt mixtures called "CMA" that can be produced at a significantly lower temperature than WMA.

## 3. Laboratory Evaluation of "Zero-M" Polymer Cool Mix Asphalt

### 3.1. Materials

As shown in Figure 3, the polymer cool mix asphalt (PCMA) additive called "Zero-M" was developed through the gelation process of finely ground SBS polymer, a thermoplastic copolymer of Styrene ($C_6H_5CH=CH_2$) and Butadiene ($CH_2=CHCH=CH_2$) along with process oil, polybutene, and adhesive resin. The process oil reduces drag force and highly reactive polybutene enhances surface activity with adhesive functionality, which is effective in improving the dispersion and fluidity of asphalt. To produce a PMA of PG 76-22, the dosage rate of 14% of the total binder content was added to PG 64-22 binder at a mixing temperature of 110 °C. Table 1 summarizes the test results of Zero-M with respect to ASTM standards.

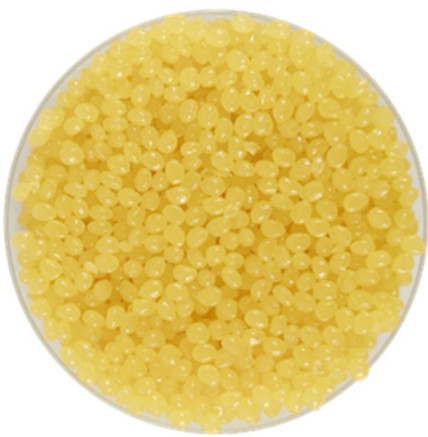

**Figure 3.** PCMA additive called "Zero-M" manufactured by Hansoo Road Industry.

**Table 1.** Test results of polymer cool mix asphalt additive.

| Parameter | Unit | Test Results | Standard |
|---|---|---|---|
| Specific Gravity | g/cc | 0.976 | ASTM D 792 [13] |
| Hardness | shore A | 10 | ASTM D 2240 [14] |
| Tensile Strength | kgf/cm$^2$ | 5 | ASTM D412 [15] |
| Elongation | % | 1282 | ASTM D 412 |
| 100% Modulus | kgf/cm$^2$ | 3 | ASTM D 412 |
| Melt Flow Index | g/10 min | 56 | ASTM D 1238 [16] |

### 3.2. Laboratory Tests

To determine PG grade, a set of SuperPave binder tests were performed on the asphalt binder of PG 62-22 (86%) with PCMA additive (14%), and the test results are summarized in Table 2. Based on the SuperPave test results in Table 2, it can be concluded that 14% PCMA additive can produce PG 72-22 asphalt at 110 °C. PCMA mixtures were then produced by mixing PCMA binder (5.2%) with 13 mm max aggregates (94.8%) as shown in Table 3. As summarized in Table 4, PCMA mixtures were tested following the standard testing protocols. As can be seen in Table 4, PCMA mixtures passed all standard tests.

**Table 2.** PCMA binder test results.

| SuperPave Test Procedure (PG 77-22) | Unit | Test Result | Criteria |
|---|---|---|---|
| Viscosity (110 °C) | Pa·s | 1.204 | ≤3.0 |
| Viscosity (135 °C) | Pa·s | 0.521 | ≤3.0 |
| G*/sin δ (76 °C, Original) | kPa | 1.401 | ≥1.0 |
| G*/sin δ (76 °C, after RTFO) | kPa | 2.89 | ≥2.2 |
| G*sin δ (31 °C, after PAV) | kPa | 2940 | ≤5000 |
| Stiffness (−12 °C) | kPa | 262 | ≤300 |
| M-value (−12 °C) | - | 0.38 | ≥0.3 |
| Flash Point | °C | 318 | ≥230 |
| Mass Loss (After RTFO) | % | 0.09 | ≤1.0 |

**Table 3.** Gradation of aggregates.

| Sieve Size (mm) | % Passing |
|---|---|
| 13 | 100 |
| 10 | 85.5 |
| 5 | 44 |
| 2.5 | 28 |
| 0.6 | 13.5 |
| 0.3 | 9.8 |
| 0.15 | 7.3 |
| 0.08 | 4.9 |

**Table 4.** PCMA mixtures test results.

| Test Procedures | Unit | Test Results | Criteria |
|---|---|---|---|
| Marshall Stability | N | 9344 | ≥6000 |
| Flow | 1/100 cm | 33.9 | 20~40 |
| Air Void | % | 4.5 | 3~5 |
| Voids Filled with Asphalt | % | 79 | 65~80 |
| Tensile Strength | N/mm$^2$ | 0.97 | ≥0.8 |
| Toughness | N.mm | 10,000 | ≥8000 |
| Dynamic Stability | times/mm | 4111 | ≥3000 |

## 4. Field Implementation of "Zero-M" Cool MIX Asphalt in Korea

*4.1. Laboratory Tests of Field Mixtures*

Based on the Marshall design method, a 5.6% total binder content was adopted (4.9% asphalt with a penetration grade of 60/80 and 0.7% Zero-M) for field application. The aggregates consisted of coarse aggregates with a maximum size of 13 mm, crushed sands as fine aggregates and a lime powder as a filler. For mixing at the asphalt plant, aggregates and asphalt were heated to 110 °C and 150 °C, respectively, whereas fillers and Zero-M modifier were added at a room temperature. The average temperature of mixtures was 106 °C. The average density and air voids of laboratory-compacted field mixtures were 2.420 (average of 2.434, 2.426 and 2.399) and 3.7%, whereas those of the control mixtures without PCMA were 2.440 (average of 2.464, 2.424 and 2.431) and 3.4%, respectively.

For laboratory tests, field mixtures were compacted at 100 °C using the Marshall hammer with 75 blows on each side. Compacted specimens were tested for Marshall stability, air voids and dynamic stability. Table 5 shows the laboratory test results of the field PCMA mixtures produced at the asphalt plant. As can be seen in Table 5, the Marshall stability was 11,200 N, which exceeded the standard of 7500 N. All other test results, including air voids, met the criteria. It should be noted that the laboratory compacted PCMA field mixtures exhibited a compaction ratio of 0.98 well above the standard limit of 0.96 in Korea.

Next, the Hamburg Wheel Tracking test was performed on field PCMA mixtures at 50 °C, and the test results are shown in Figure 4 [17]. It is interesting to note that the field PCMA mixtures did not exhibit a stripping point during 20,000 repeated load cycles, which indicates an excellent resistance to moisture damage. Further, the average rut depth was less than 5 mm after 20,000 load cycles, which confirms a good resistance to rutting.

**Table 5.** Laboratory test results of PCMA mixtures produced at the asphalt plant.

| Test Procedures | Unit | Test Results | Criteria |
|---|---|---|---|
| Mix temperature | °C | 106 | 100–120 |
| Marshall Stability | N | 11,200 | ≥7500 |
| Flow | 1/100 cm | 35 | 20~40 |
| Air Void | % | 3.7 | 3~5 |
| Voids Filled with Asphalt | % | 78 | 65~80 |
| Tensile Strength | N/mm$^2$ | 0.92 | ≥0.8 |
| Dynamic Stability | times/mm | 5250 | ≥3000 |
| Compaction Ratio | - | 0.98 | ≥0.96 |

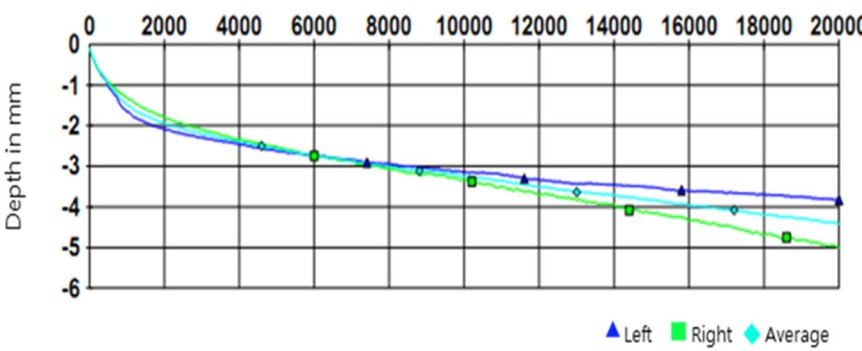

**Figure 4.** HWT test result of PCMA produced at asphalt plant in Jeju, Korea.

*4.2. Field Implementation Results*

On 13 October 2023, the "Zero-M" PCMA mixtures were produced at the Seoil Asphalt Plant shown in Figure 5, which is located 30 min from the 150 m-long test section in Seogwipo city, Jeju Island, Korea. Due to the PCMA's low aggregate temperature of 110 °C, compared to HMA, the energy consumption at the asphalt plant using LPG as a fuel decreased by 60%, from 7 L/ton to 2.8 L/ton, and carbon dioxide emission decreased by 63% compared to HMA.

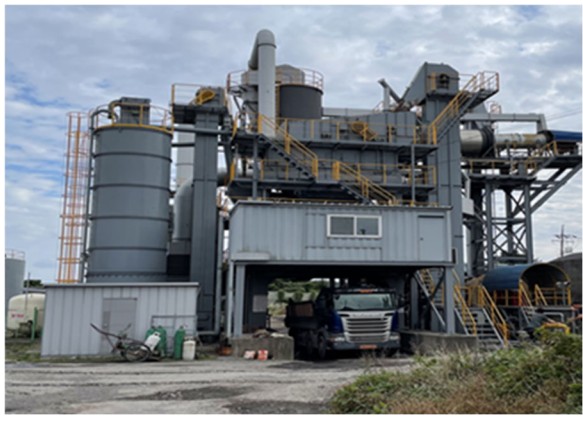

**Figure 5.** Seoil asphalt plant to produce Zero-M mixtures in Seogwipo City, Korea.

As shown in Figure 6a, the severely distressed road with significant cracking under a high traffic volume in Seogwipo City was rehabilitated using the "Zero-M" PCMA. As shown in Figure 6, the existing pavement was milled to a depth of 5 cm, and a 5 cm overlay of PCMA was applied, which was then compacted by macadam roller followed by tire and tandem rollers.

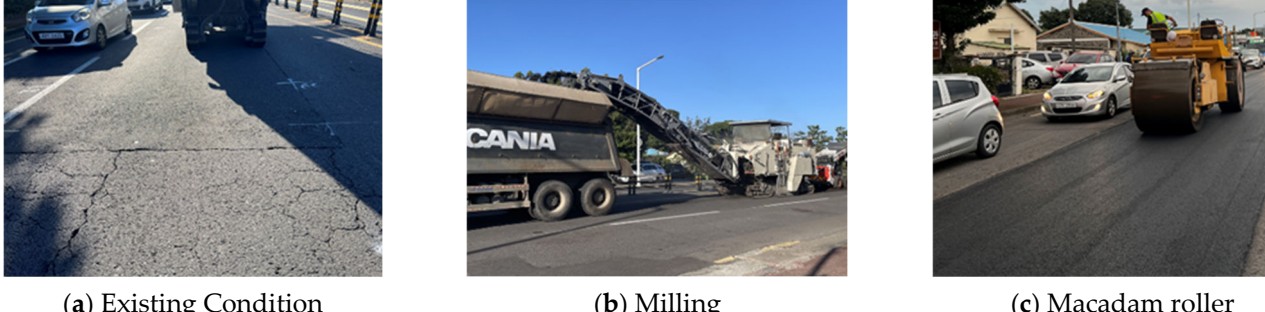

(**a**) Existing Condition      (**b**) Milling      (**c**) Macadam roller

**Figure 6.** Construction of 150 m-long "Zero-M" PCMA test section in Seogwipo City, Korea.

The temperatures of placement and initial compaction were 104 °C and 95 °C, respectively. Densities and air voids were measured from the cores extracted from pavements after construction. Average density and air voids of field PCMA cores were 2.357 (average of 2.362 and 2.352) and 4.6%, whereas the those of the control section were 2.386 (average of 2.399 and 2.372) and 4.2%, respectively.

## 5. Field Implementation of "Zero-M" Cool MIX Asphalt in Italy

The PCMA material called "Zero-M" was added to Bitumen 50/70 to produce CMA mixtures at Bindi Inc.'s batch plant in Florence, Italy. The types of materials along with their temperatures added to the pugmill are illustrated in Figure 7.

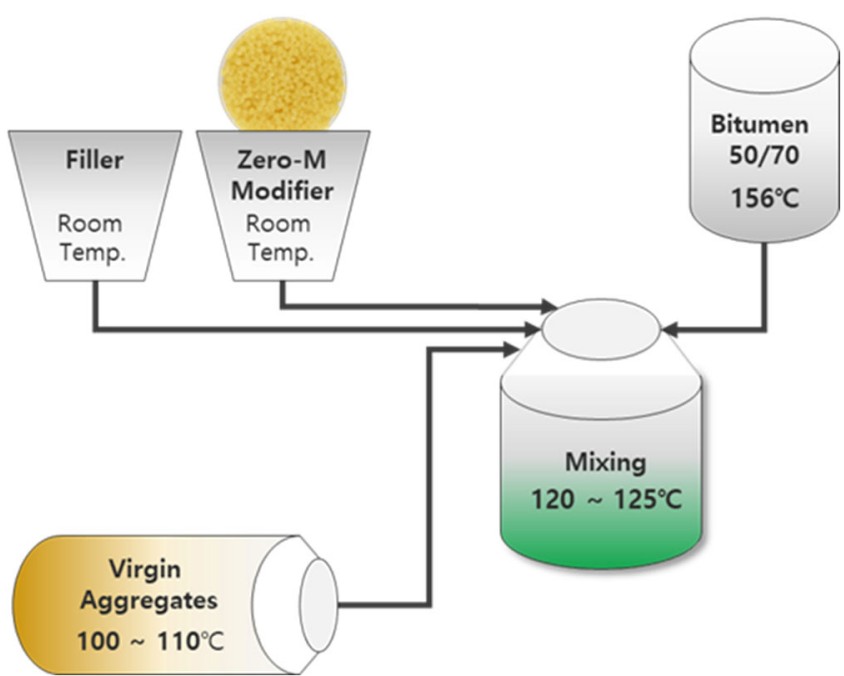

**Figure 7.** Mixing process by Bindi, Inc. for building test road construction in Florence, Italy.

When 30% RAP materials are inserted into the mixer, to allow optimal mixing of HMA, the virgin aggregates are heated in the dryer oven at higher temperatures. As shown in Figure 8, the graph shows different temperatures of the virgin aggregates at temperatures between 250 and 300 °C and between 100 and 130 °C during the HMA and PCMA production processes, respectively. As can be seen from Figure 8, the temperature of aggregates used for the PCMA layer is significantly lower than temperature of aggregates used for the HMA layer.

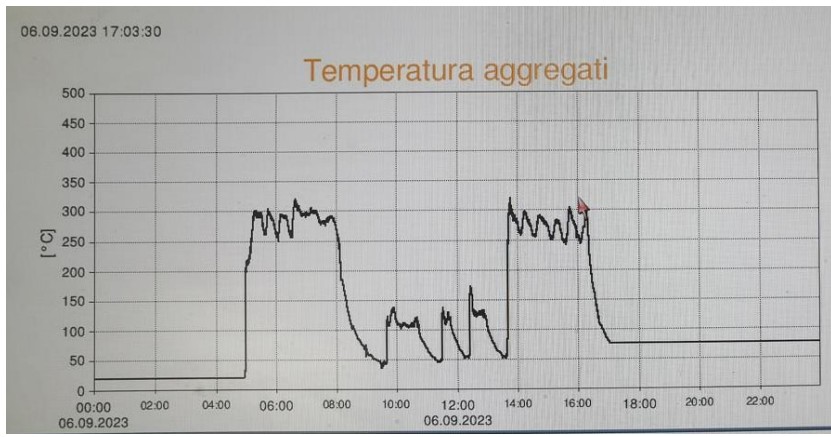

**Figure 8.** Aggregate temperatures of HMA vs. PCMA at Bindi's Batch Plant.

A 4 cm-thick PCMA surface layer was successfully constructed on top of a 6 cm HMA binder layer on a 200 m-long test section of a two-lane road of SP 16, #14 in Florence, Italy. Macadam roller was applied on the PCMA surface layer at approximately 100 °C, and a tire roller was applied at 90 °C, and then a tandem roller was applied at 70 °C. As can be seen from Figure 9, a PCMA surface layer, as shown in (a), was not collapsed by the heavy tandem roller riding on the edge of the PCMA surface layer, as shown in (b). Cores were taken from the test section to measure the density. It should be noted that the surface of PCMA core was significantly darker than that of HMA core. It can be postulated that, during the coring process, PCMA binder smeared across the aggregate surface because of a high stickiness of PCMA binder.

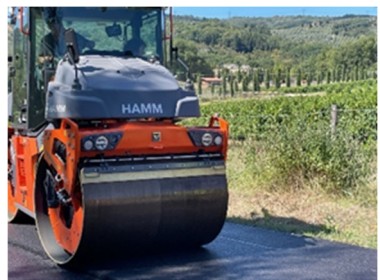
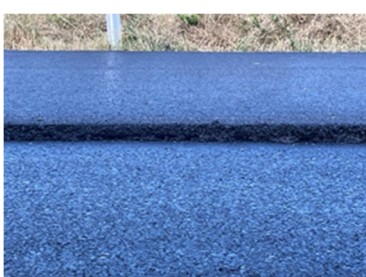
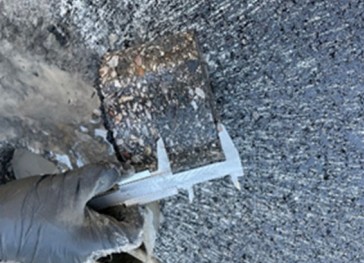

(**a**) Tandem roller on the edge     (**b**) Edge of PCMA after roller     (**c**) Core of PCMA over HMA

**Figure 9.** Pictures of compaction process of PCMA surface and PCMA/HMA core.

Average density and air voids of field PCMA mixtures were 2.341 (average of 2.338 and 2.344) and 2.66% (average of 2.79% and 2.53%), whereas the those of the control section without PCMA were 2.269 (average of 2.270 and 2.268) and 5.67% (average of 5.63% and 5.71%), respectively.

## 6. Field Implementation of "Zero-M" Cool Mix Asphalt in Vietnam

### 6.1. Laboratory Tests of Field Mixtures

The loose mixtures were collected from the paver, and then they were reheated in the laboratory to test their properties. As shown in Table 6, the laboratory-compacted field mixtures passed all criteria of Vietnam [18]. As can be seen in Table 6, as expected, air voids of field mixtures with 13% Zero-M additive were at 4.5% which is lower than 5.5% with 10% Zero-M additive. However, the Marshall Stability of 13% Zero-M additive was lower, but the flow number was higher than mixtures with 10% Zero-M additive.

**Table 6.** Properties of PCMA.

| Properties | Testing Result | | Criteria |
|---|---|---|---|
| | **10% Zero-M** | **13% Zero-M** | |
| Laboratory Bulk Density | 2.548 | 2.549 | - |
| Theoretical Max. Density | 2.683 | 2.678 | - |
| Air voids, % | 5.5 | 4.5 | 3–6 |
| Voids in the mineral aggregate % | 15.9 | 15.0 | Min 13 |
| Marshall Stability, kN | 10.85 | 9.53 | Min 8.0 |
| Marshall flow, mm | 3.34 | 3.69 | 2.0–4.0 |
| Retained Marshall Stability, % | 93 | 94 | Min 85 |
| Coatability Index, % | 100 | 100 | Min 95 |

The Hamburg wheel tracking test was performed on the laboratory-compacted field mixtures following AASHTO T 324. As shown in Figure 10, the field mixtures exhibited a good rutting resistance at 20,000 repetitions. Compared to the 13% Zero-M sample, the rutting of 10% Zero-M sample was lower by 16.1% at 20,000 repetitions.

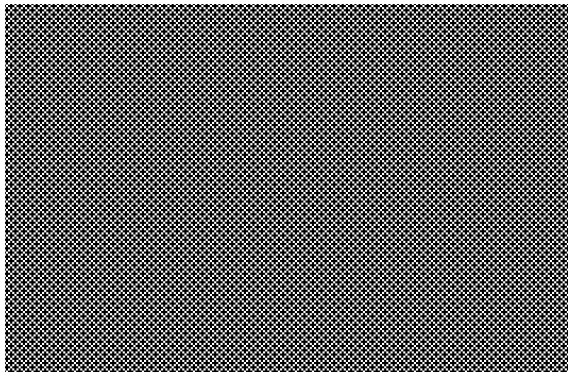

**Figure 10.** Hamburg wheel tracking test results of mixtures with 10% and 13% Zero-M.

Following ASTM D8225 [19], the Indirect Tensile Cracking Test (IDEAL-CT) was performed on the laboratory-compacted field mixtures. As shown in Figure 11, the $CT_{Index}$ of 13% Zero-M additive was higher than mixtures with 10% Zero-M additive, which indicates that an additional 3% Zero-M amount increased cracking resistance by 42.4%.

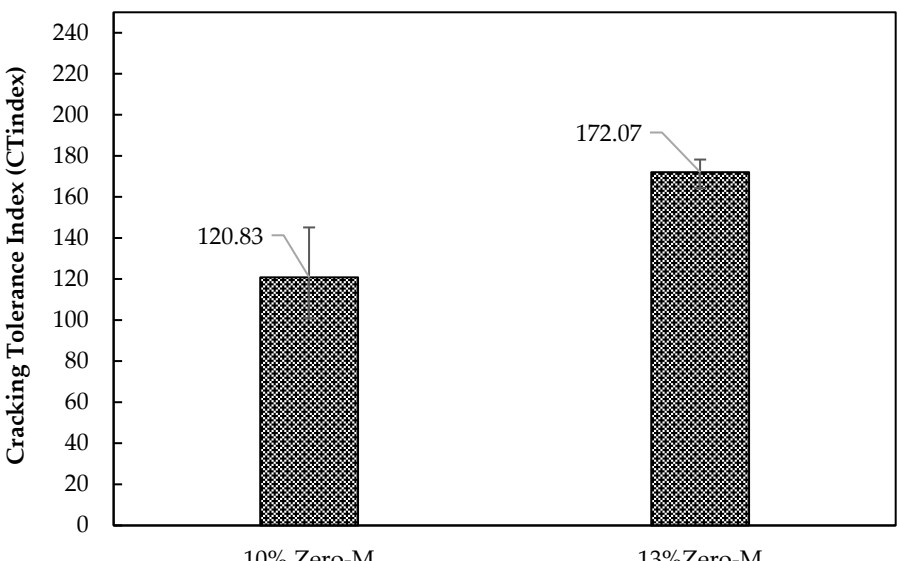

**Figure 11.** Indirect Tensile Cracking Test (IDEAL-CT) results of laboratory-compacted field mixtures with 10% and 13% Zero-M additives.

### 6.2. Field Implementation Results

The 5 cm-thick "Zero-M" PCMA surface layers (105 m$^2$ each with 10% and 13% "Zero-M" additives) were successfully constructed on top of an existing 8 cm AC-19 HMA binder layer and 12 cm ATB 25 asphalt-treated base on 965 Road in Ba Ria-Vung Tau Province, Vietnam. Figure 12 shows the asphalt plant and PCMA mix loading in a truck. The Zero-M PCMA mixtures were produced in asphalt plant with an aggregate temperature of 100 °C and the asphalt binder temperature of 150 °C, resulting in a mix temperature of 115 °C.

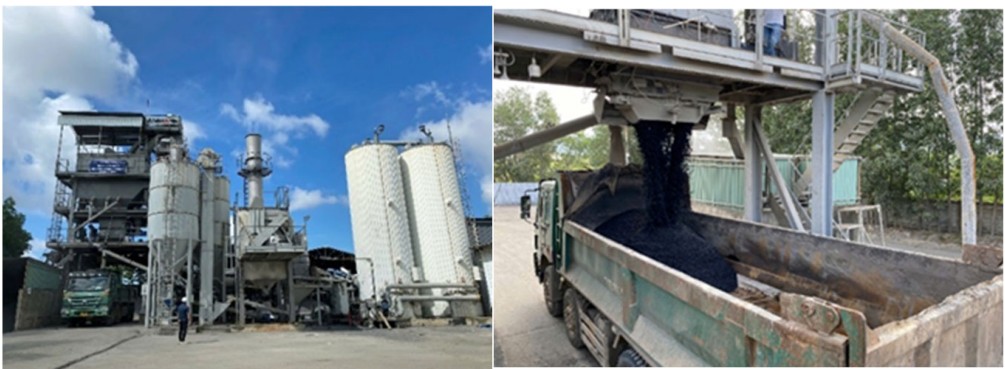

(**a**) Asphalt Batch Mixing Plant          (**b**) Loading PCMA mix to a truck

**Figure 12.** Asphalt batch plant and loading of PCMA mixtures in Vietnam.

As shown in Figure 13, Zero-M PCMA mixtures were successfully placed at 115 °C and compacted at temperatures between 100 and 105 °C.

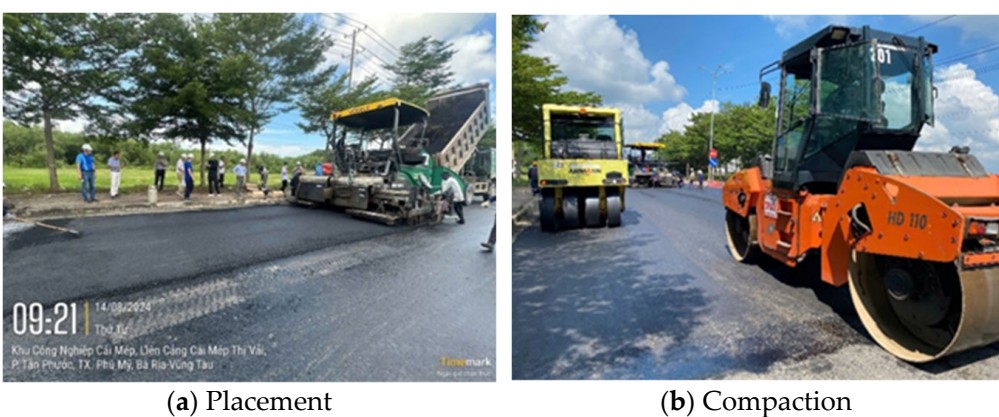

(**a**) Placement          (**b**) Compaction

**Figure 13.** (**a**) Placement and (**b**) Compaction of Zero-M PCMA Mixtures.

As summarized in Table 7, cores taken from the construction site were tested for their Marshall stability, rut depth, field densities and air voids. Compared to the results of the laboratory-compacted field mixtures, Marshall stability value of cores was lower by 26.3%, and the rut depth was lower by 43.1%. Average density and air voids of field PCMA mixtures with 10% Zero-M additive were 2.526 (average of 2.526, 2.536, and 2.516) and 5.85% and PCMA mixtures with 13% Zero-M additive were 2.530 (average of 2.542, 2.540 and 2.510) and 5.53%, respectively, whereas the those of the control Polymer-Modified Asphalt (PMA) section without Zero-M were 2.528 (average of 2.518, 2.540 and 2.527) and 5.81%, respectively. It should be noted that although the Zero-M mixtures were compacted at significantly lower temperature than PMA by 65 °C, field densities and air voids of Zero-M PCMA cores were quite similar to those of PMA, while meeting the density and air voids criteria in Vietnam.

**Table 7.** Core sample testing results.

| Properties | Testing Result | | | Criteria |
|---|---|---|---|---|
| | **10% Zero-M** | **13% Zero-M** | **PMA** | |
| In-Place Density | 2.526 | 2.530 | 2.515 | - |
| Theoretical Max. Density | 2.683 | 2.678 | 2.666 | - |
| Laboratory Bulk Density | 2.548 | 2.549 | 2.529 | - |
| % of Theoretical Max. Density | 94.1 | 94.5 | 94.3 | 92–97 |
| Air Voids, % | 5.85 | 5.53 | 5.66 | - |
| Marshall Stability, kN | 7.8 | 7.2 | - | - |
| Rutting dept, mm | 4.83 | 5.95 | - | Max12.5 |

## 7. Summary and Conclusions

Although Warm Mix Asphalt (WMA) has been widely used in the US and the world, its usage has been limited due to an inconsistent definition of WMA. For example, NAPA defines WMA as "technologies that allow the production of asphalt pavement material to lower the temperature at which the material is mixed and placed on the road by 10 to 100 degrees F (6 to 38 °C)". The temperature reduction range is not only too wide but also the minimum temperature threshold value of 10 °F (6 °C) to be classified as WMA is too small. In this paper, a new category of Cool Mix Asphalt (CMA) is presented to distinguish it from the WMA based on the actual production temperatures not by the reduction temperature from that of HMA. It is proposed that HMA should be defined as asphalt mixtures produced at temperatures between 140 and 160 °C (between 284 and 320 °F), WMA at temperatures between 120 and 140 °C (between 248 and 284 °F), and CMA at temperatures between 100 and 120 °C (212 to 248 °F). These clearer definitions being adopted by the world will allow researchers, engineers and contractors to compare apples with apples rather than apples and oranges.

In this paper, a new polymer cool mix asphalt (PCMA) additive called "Zero-M" is presented along with its lab test results and its successful field implementation experiences in Korea, Italy and Vietnam. It was observed that the surface of PCMA cores appeared darker than that of HMA cores due to a high stickiness of Zero-M additive. It should be noted that, particularly during compaction process in Italy, the PCMA surface layer did not collapse despite a heavy tandem roller riding on the edge of the PCMA surface layer.

The average air void values from the cores extracted from both PCMA and control HMA test sections constructed in three different countries are summarized in Table 8. As can be seen in Table 8, air voids of PCMA cores in Korea and Vietnam were slightly higher than those of the control HMA/PMA cores, whereas air voids of PCMA cores in Italy were significantly lower than those of control HMA cores.

**Table 8.** Air voids of PCMA and control HMA cores in Korea, Italy and Vietnam.

| Test Section\Country | Korea | Italy | Vietnam |
|---|---|---|---|
| PCMA | 4.6% | 2.66% | 5.85%/5.53% * |
| HMA/PMA ** | 4.2% | 5.67% | 5.81% |

\* Air voids of PCMA cores with 10% and 13% Zero-M additives. ** HMA for the control test section in Vietnam is polymer-modified asphalt.

In the future, average temperature values of HMA, WMA and CMA along with standard deviation values should be considered for developing a performance-based incentive program such that projects with lower average temperature values with small standard deviation values are rewarded with a bonus in each category of HMA, WMA and

CMA because a small standard deviation value of multiple temperature measurements indicates a consistent quality of asphalt materials during the production process. The recyclability and durability of CMA should be studied through laboratory mechanical tests in the future.

**Author Contributions:** The authors confirm contribution to the paper as follows: study conception and design: H.L. and Y.-i.K.; data collection: L.N.N., E.S. and Y.-i.K.; analysis and interpretation of results: H.L., L.N.N. and E.S.; draft manuscript preparation: H.L. and L.N.N. All authors have read and agreed to the published version of the manuscript.

**Funding:** This research received external funding support from JEJU TECHNOPARK.

**Data Availability Statement:** The raw data supporting the conclusions of this article will be made available by the authors on request.

**Acknowledgments:** The authors would like to thank Cesare Sangiorgi at University of Bologna for measuring densities and air voids of cores taken from the test section in Italy.

**Conflicts of Interest:** Author Elena Sturlini was employed by the company Bindi S.P.A. Author Young-ik Kim was employed by the company Hansoo Natech Co., Ltd. The remaining authors declare that the research was conducted in the absence of any commercial or financial relationships that could be construed as a potential conflict of interest.

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
