# Peer review of "Cool Mix Asphalt—Redefining Warm Mix Asphalt with Implementations in Korea, Italy and Vietnam"

_infrastructures, doi:10.3390/infrastructures10010024_

Round 1

Reviewer 1 Report

Comments and Suggestions for Authors

This paper broadly discusses WMA, HMA, and CMA. It also introduces a new PCMA additive called “Zero-M.” This paper needs considerable revision because the overall presentation of the paper is messy and it is not professionally written.

1.     The purpose is not clear. Do authors want to compare the appropriateness/inappropriateness of defining asphalt mixtures by production temperature and reduction temperature? Or “Do they want to evaluate the performance of “Zero-M?”

2.     Abbreviations are not well defined. Ex: NOVA (line 49), PG (line 59), SBS (line 123), DOT (line 68). Abbreviations should be explained at the first appearance. It is not right to evaluate the paper assuming the meaning of these abbreviations.

3.     There are careless mistakes. Ex: 2,424 (line 151).

4.     Please give the details of standard specifications in the references. Ex: Hamburg Wheel Tracking test. What kind of specification is this?

5.     What is the chemical formula of SBS polymer? What is the mechanism of “Zero-M” that reduces the mixing temperature?

6.     What is the meaning of the sentence: “By defining their actual……….(lines 271-273).

7.     What is the meaning of the sentence: “with small standard deviation values are rewarded with bonus in each category of……. (lines 293-294).

Comments on the Quality of English Language

Authors should improve the readability of the paper. 

Author Response

Comment 1: The purpose is not clear. Do authors want to compare the appropriateness/inappropriateness of defining asphalt mixtures by production temperature and reduction temperature? Or “Do they want to evaluate the performance of “Zero-M?”

Response 1: To clarify the purpose of the paper, The following paragraph was added in lines 64-67 at the end of the Introduction Section.

The main purposes of this paper are two folds: 1) clearly defining HMA, WMA and “Cool Mix Asphalt (CMA)” by their absolute production temperature ranges and 2) demonstrating the successful field implementations of “Polymer Cool Mix Asphalt (PCMA)” using a “Zero-M” additive in Korea, Italy and Vietnam.

Comment 2: Abbreviations are not well defined. Ex: NOVA (line 49), PG (line 59), SBS (line 123), DOT (line 68). Abbreviations should be explained at the first appearance. It is not right to evaluate the paper assuming the meaning of these abbreviations.

Response 2. NOVA is not an acronym but a unique name for the award that is given by Construction Industry Forum.  Authors Defined "Performance Grade (PG)" and European Asphalt Pavement Association (EAPA) on line 59, Iowa Department of Transportation (IDOT) on line 69, and  Styrene-Butadiene-Styrene (SBS) on line 123.

Comment 3: There are careless mistakes. Ex: 2,424 (line 151).

Response 3: Corrected to 2.424 on line 152

Comment 4: Please give the details of standard specifications in the references. Ex: Hamburg Wheel Tracking test. What kind of specification is this?

Response 4: AASHTO T-324 specification detail was added in the reference on line 308-309.

Comment 5. What is the chemical formula of SBS polymer? What is the mechanism of “Zero-M” that reduces the mixing temperature?

Response 5: chemical formula of SBS was added on lines 123-124. a thermoplastic copolymer of Styrene (C6H5CH=CH2) and Butadiene (CH2=CHCH=CH2 ) and the following sentences were added on lines 123-127.

a thermoplastic copolymer of Styrene (C6H5CH=CH2) and Butadiene (CH2=CHCH=CH2) along with process oil, polybutene, and adhesive resin. The process oil reduces a drag force and highly reactive polybutene enhances surface activity with adhesive functionality, which is effective in improving the dispersion and fluidity of asphalt.

 Comment 6. What is the meaning of the sentence: “By defining their actual……….(lines 271-273).

Response 6.  This statement is confusing and therefore this sentence was removed from the paper.

Comment 7. What is the meaning of the sentence: “with small standard deviation values are rewarded with bonus in each category of……. (lines 293-294).

response 7.  the following sentence was added  "because a small standard deviation value of multiple temperature measurements indicates a consistent quality of asphalt materials during the production process." on lines 299-301.

Reviewer 2 Report

Comments and Suggestions for Authors

1. Generally this research is interesting. It introduces a cool mix asphalt technology which can reduce the temperature for the construction of asphalt pavement. Technically, I don't really believe it is cool mix, as the temperature is still more than 100 oC. Probably the authors can state it as enhanced or improved warm mix technology.

2. The introduction section is weak. Authors just introduced some history of WMA, not technical stuff. It is needed to intorduce more technical stuff like mechanical properties, or rheological properties of WMA. Please refer to the following literatures to improve this section, such as the fatigue properties of asphalt binders being used for asphalt pavement.

https://doi.org/10.1016/j.conbuildmat.2024.136698

3. It is suggested to add the statistic data for Asia and Oceania, as these parts are also very important and have significant shares of WMA.

4. Have you assessed the recyclibility of your "cool mix"? Please discuss a little bit about the recyclibility, some references for you:

5. Also, the durability is critical for the asphalt pavements. Please refer to the following paper and discuss a little bit about the potential durability of PCMA.

6.It is suggested to perform more mechanical tests in your future study.

Author Response

Comment 1. Generally this research is interesting. It introduces a cool mix asphalt technology which can reduce the temperature for the construction of asphalt pavement. Technically, I don't really believe it is cool mix, as the temperature is still more than 100 oC. Probably the authors can state it as enhanced or improved warm mix technology.

Response 1. Authors would like to create a distinguishable term of CMA from WMA. we added the following paragraph on lines 131-139 to reiterate our position.

It is proposed in this paper that HMA should be defined as asphalt mixtures produced at temperatures between 140 and 160°C (between 284 and 320°F), WMA as production temperatures between 120 and 140°C (between 248 and 284°F), and CMA as production temperatures between 100 and 120°C (212 to 248°F). It may be argued that temperatures between 100 and 120°C are not a cool temperature but the same argument can be made for WMA that temperatures between 120 and 140°C (or a production temperature reduction of just 6°C from HMA) are not warm either but it has been already accepted as an industry standard. Therefore, it is necessary to create a new category of asphalt mixtures called “CMA” that can be produced at a significantly lower temperature than WMA.

2. The introduction section is weak. Authors just introduced some history of WMA, not technical stuff. It is needed to intorduce more technical stuff like mechanical properties, or rheological properties of WMA. Please refer to the following literatures to improve this section, such as the fatigue properties of asphalt binders being used for asphalt pavement.

https://doi.org/10.1016/j.conbuildmat.2024.136698

Thank you for your suggestion and the paper on fatigue properties of aged asphalt binder.  I added three relevant references in the paper with a focus on production and compaction temperatures because this paper does not address laboratory mechanical properties and rheological properties.  The purpose of our paper is added in the introduction section (please see the text in red on lines 60-78)  

3. It is suggested to add the statistic data for Asia and Oceania, as these parts are also very important and have significant shares of WMA.

Additional references from Vietnam, Korea and Middle East were added on lines 60 - 72 in the introduction section.

4. Have you assessed the recyclibility of your "cool mix"? Please discuss a little bit about the recyclibility, some references for you:

Great suggestion.  We have not considered a recyclability of CMA for this paper. Therefore, we added a statement in the summary and conclusions as a future studies. on lines 325-327.

5. Also, the durability is critical for the asphalt pavements. Please refer to the following paper and discuss a little bit about the potential durability of PCMA.

Great suggestion.  We have not considered a durability of CMA for this paper. Therefore, we added a statement in the summary and conclusions as a future studies. on lines 325-327.

6.It is suggested to perform more mechanical tests in your future study.

Authors agree.  We are currently performing laboratory tests of PCMA mixtures with different amounts of RAP materials and we would like to present out results on mechanical tests in the future.